# Fetal Aortic Blood Flow Velocity and Power Doppler Profiles in the First Trimester: A Comprehensive Study Using High-Definition Flow Imaging

**DOI:** 10.3390/bioengineering11040378

**Published:** 2024-04-15

**Authors:** Yi-Cheng Wu, Ching-Hsuan Chen, Hsin-Tzu Lu, Yu-Li Lee, Pi-Yu Chen, Ting-Yu Wu, Ming-Hsun Tien, Chiung-Hui Wu, Jack Yu-Jen Huang, Ching-Hua Hsiao, Woei-Chyn Chu

**Affiliations:** 1Department of Biomedical Engineering, National Yang-Ming Chiao-Tung University, Taipei 112304, Taiwan; yicheng97.y@nycu.edu.tw (Y.-C.W.); karenlu.y@nycu.edu.tw (H.-T.L.); 2Department of Obstetrics and Gynecology, Ton Yen General Hospital, Hsinchu 302048, Taiwan; mhtien77@gmail.com (M.-H.T.); chiunghuiwu@yahoo.com.tw (C.-H.W.); jyjhuang@stanford.edu (J.Y.-J.H.); 3Taiwan IVF Group Center for Reproductive Medicine & Infertility, Hsinchu 302053, Taiwan; piyu713@gmail.com (P.-Y.C.); a9518348@yahoo.com.tw (T.-Y.W.); 4Department of Obstetrics and Gynecology, Fuyou Branch, Taipei City Hospital, Taipei 100027, Taiwan; chchen@tpech.gov.tw (C.-H.C.); b1303@tpech.gov.tw (Y.-L.L.)

**Keywords:** power Doppler ultrasound, aortic isthmus, high-definition flow imaging (HDFI), isthmic flow index, isthmic systolic index, first-trimester screening, fetus

## Abstract

Objectives: This study aimed to establish reference values for fetal aortic isthmus blood flow velocity and associated indices during the first trimester, utilizing a novel ultrasonographic technique known as high-definition flow imaging (HDFI). Additionally, the correlation between Doppler profiles of aortic blood flow and key fetal parameters, including nuchal thickness (NT), crown-rump length (CRL), and fetal heartbeat (FHB), was investigated. Methods: A total of 262 fetuses were included in the analysis between December 2022 and December 2023. Utilizing 2D power Doppler ultrasound images, aortic blood flow parameters were assessed, including aortic peak systolic velocity (PS), aortic end-diastolic velocity (ED), aortic time average maximal velocity (TAMV), and various indices such as aortic systolic velocity/diastolic velocity (S/D), aortic pulsatile index (PI), aortic resistance index (RI), aortic isthmus flow velocity index (IFI), and aortic isthmic systolic index (ISI). Concurrently, fetal FHB, NT, and CRL were evaluated during early trimester Down syndrome screening. Results: Significant findings include a positive correlation between gestational age (GA) and PS (PS = 3.75 × (GA) − 15.4, r^2^ = 0.13, *p* < 0.01), ED (ED = 0.42 × (GA) − 0.61, r^2^ = 0.04, *p* < 0.01), PI (PI = 0.07 × (GA) + 1.03, r^2^ = 0.04, *p* < 0.01), and TAMV (TAMV = 1.23 × (GA) − 1.66, r^2^ = 0.08, *p* < 0.01). In contrast, aortic ISI demonstrated a significant decrease (ISI = −0.03 × (GA) + 0.57, r^2^ = 0.05, *p* < 0.05) with gestational age. No significant correlation was observed for aortic RI (*p* = 0.33), S/D (*p* = 0.39), and IFI (*p* = 0.29) with gestational age. Aortic PS exhibited positive correlations with NT (0.217, *p* = 0.001) and CRL (0.360, *p* = 0.000) but a negative correlation with FHB (−0.214, *p* = 0.001). Aortic PI demonstrated positive correlations with CRL (0.208, *p* = 0.001) and negative correlations with FHB (−0.176, *p* = 0.005). Aortic TAMV showed positive correlations with NT (0.233, *p* = 0.000) and CRL (0.290, *p* = 0.000) while exhibiting a negative correlation with FHB (−0.141, *p* = 0.026). Aortic ISI demonstrated negative correlations with NT (−0.128, *p* = 0.045) and CRL (−0.218, *p* = 0.001) but a positive correlation with FHB (0.163, *p* = 0.010). Conclusions: Power Doppler angiography with Doppler ultrasound demonstrates the ability to establish accurate reference values for fetal aortic blood flow during the first trimester of pregnancy. Notably, aortic PS, TAMV, and ISI exhibit significant correlations with NT, CRL, and FHB, with ISI appearing more relevant than IFI, PS, TAMV, and FHB. The utilization of HDFI technology proves advantageous in efficiently detecting the site of the aortic isthmus compared to traditional color Doppler mode in early second trimesters.

## 1. Introduction

The aortic isthmus (AI) is a pivotal segment of the aorta, situated between the origin of the left subclavian artery and the connection of the ductus arteriosus to the descending aorta. In prenatal life, it plays a crucial role in balancing the brachiocephalic circulation, supplying the upper body (including the brain), and the subdiaphragmatic circulation, providing the lower body and placenta [1,2,3]. While Doppler flow velocity waveforms can be acquired using B-mode imaging and pulsed-wave Doppler, the optimal identification of vessels and showcasing the direction of blood flow is achieved through color-directed pulsed-wave Doppler interrogation. This technique allows for precise cursor positioning [4,5]. The traditional longitudinal aortic arch view (LAA view) is favored for its ease in visualizing the origin of the left subclavian artery [6]. Despite numerous studies focusing on assessing the aortic isthmus in the second trimester [6,7,8], limited research has yet to be conducted in the first trimester. This underscores the importance of further exploration during the early stages of pregnancy to enhance our understanding of the aortic isthmus’s dynamics and potential implications for prenatal care.

As we know, the capability to detect blood flow velocity in the AI using color Doppler ultrasound typically initiates around 12 to 14 weeks of gestation. This timing aligns with the gradual maturation of the fetal heart and circulatory system during the early weeks of pregnancy, eventually reaching a point where they can be reliably visualized and measured using ultrasound techniques. However, the exact timing for detecting blood flow velocity in the aortic isthmus may vary depending on the sonographer’s skill, ultrasound equipment, and fetal position.

A notable advancement in this field is HDFI from GE Voluson E6 and E8, incorporating bi-directional power Doppler imaging. This technology has proven to be a valuable tool by combining the sensitivity of Power Doppler with directional flow information. Consequently, it enhances the visualization of fine details and improves blood flow sensitivity within the specific anatomy of interest [9,10,11,12,13,14,15,16]. The utilization of HDFI contributes to a more comprehensive and accurate assessment of blood flow dynamics in the AI during the early stages of gestation.

This research employs a novel ultrasound technique (HDFI) to establish standard reference values for fetal aortic isthmus blood flow velocity and indices in the first trimester. Our secondary goal is to explore the relationship between Doppler indices of blood flow in the fetal AI and critical parameters such as nuchal thickness (NT), crown-rump length (CRL), and fetal heartbeat (FHB).

## 2. Materials and Methods

### 2.1. Study Design and Patient Population

This study was approved by the Medical Ethics and Institutional Review Board of the Taoyuan General Hospital, Ministry of Health and Welfare (IRB No: TYGH 112002). This is a retrospective, cross-sectional study that involved 262 fetuses with gestational ages ranging from 11 to 14 completed weeks (Figure 1). The participants (252 fetuses) were enrolled at the ultrasound center of Taiwan IVF Group between December 2022 and December 2023. The inclusion criteria for the study were as follows: (1) Outpatients with gestational age confirmed by sonography in the first trimester who underwent first trimester Down syndrome screening with ultrasound examination. (2) Normal fetuses, defined as those without structural malformations and chromosome anomalies, are usually at rest. (3) Normal fetal aorta that could be visualized under power Doppler mode. (4) The fetal aorta could be assessed with color Doppler ultrasound for blood flow velocity and index. The exclusion criteria included: (1) Twin pregnancy. (2) Fetuses with congenital cardiovascular or other congenital anomalies. (3) Non-standard measurement of fetal CRL. (4) Non-quiescent fetuses.

### 2.2. Power Doppler Angiography of Transabdominal Ultrasound (PDA-TA)

All 252 normal singleton fetuses were scanned using an ultrasound system (GE, Voluson TM E6, and E8) to assess the fetal aorta. Initially, we examined the fetal heart through the pregnant woman’s abdomen using a 2D ultrasound probe (with C1-5-D) in grayscale mode. First, we obtained a clear longitudinal section of the fetal aortic arch while measuring the fetal CRL in a lying-up posture. Then, we pressed the power Doppler mode to make the signal appear on the entire aorta while waiting for the fetus to be entirely still. Then, we pressed Doppler mode (as shown in Figure 1) and placed the sample volume in an optimal location adjacent to the access to the aortic arch. The sample volume gap was kept less than or equal to 1 mm. The angle between the Doppler vernier and the assumed direction of blood flow was always kept at 0 degrees. After obtaining at least three consecutive uniform waveforms, we froze the images and stored them for subsequent data analysis. We executed and saved all records, ensuring that Doppler ultrasound waveforms of the aorta were obtained in the same fetal position and posture.

### 2.3. Assessment of Blood Flow Velocity Waveforms and Indices within the Fetal Aorta

Following adjustments to the power Doppler Angio mode (PRF: 1.8 kHz, WMF: low 1), detectable Doppler signals, defined as reproducible, similar arterial waveforms obtained for at least three consecutive cardiac cycles, were analyzed. Measurements, including peak systolic velocity (PS), peak systolic velocity/end-diastolic velocity (S/D), time-averaged maximum velocity (TAMV), pulsatility index (PI), and resistance index (RI), were calculated manually (Figure 2). Additionally, Isthmus flow velocity index (IFI) = PS + ED/PS and Isthmic systolic index (ISI) = Ns (systolic nadir)/PS, were also assessed, respectively. One author (Y.-C.W.) performed all Doppler examinations to ensure consistency and to avoid interobserver variation.

### 2.4. Statistical Analysis

The reference ranges for each ultrasound index were determined using Royston’s method [17], which accounts for the monotonic change in variance with gestational age, unlike assuming a constant. The following steps were taken for each ultrasound measurement: (1) A linear regression model was separately fitted to the standard deviation for each ultrasound measurement. (2) The confidence interval for each ultrasound measurement was calculated using the standard normal distribution. (3) Pearson’s correlation coefficient was utilized to assess the correlation between different ultrasound measurements. (4) IBM SPSS 22.0 for Windows (SPSS Inc., Chicago, IL, USA) was employed for all statistical analyses. (5) A significance level of 0.05 was chosen to determine statistical significance in all analyses.

## 3. Results

### 3.1. Basic Data of the General Populations

In the analyzed population of 252 individuals, the mean maternal age was 29.2 years, with a range spanning from 19 to 40 years old. The participants’ gestational age was around 12^+4^ weeks, ranging from 11 to 14^+1^ weeks. These details are summarized in Table 1.

### 3.2. Fetal Aortic Artery Doppler Indices

Reference curves for aortic Doppler velocities and indices were visually depicted (Figure 3).

The fetal aortic artery Doppler indices were analyzed across percentiles, providing a comprehensive overview. The mean aortic PS was 31.88 ± 6.04 cm/s. Specifically, the 5th and 10th percentiles for aortic PS were 21.88 cm/s and 23.66 cm/s, respectively. On the higher end, the 90th and 95th percentiles were 40.22 cm/s and 41.89 cm/s. Similarly, the mean aortic ED was 4.66 ± 1.20 cm/s. The 5th and 10th percentiles for aortic ED were 2.80 cm/s and 3.12 cm/s, while the 90th and 95th percentiles were 6.35 cm/s and 6.88 cm/s. The mean of the aortic TAMV was 13.79 ± 2.59 cm/s. The 5th and 10th percentiles were 9.57 cm/s and 10.63 cm/s, while the 90th and 95th percentiles were 16.95 cm/s and 17.84 cm/s. The aortic S/D had a mean of 7.23 ± 2.20, with the 5th and 10th percentiles at 4.60 and 4.89 and the 90th and 95th percentiles at 9.83 and 11.59. The aortic PI had a mean of 1.97 ± 0.37, with the 5th and 10th percentiles at 1.63 and 1.69 and the 90th and 95th percentiles at 2.26 and 2.33. The aortic RI had a mean of 0.86 ± 0.08, with the 5th and 10th percentiles at 0.78 and 0.80 and the 90th and 95th percentiles at 0.90 and 0.91.

### 3.3. Additional Fetal Aortic Artery Doppler Indices (IFI and ISI)

Mean Aortic IFI: 1.15 ± 0.04, with the 5th and 10th percentiles at 1.09 and 1.10 and the 90th and 95th percentiles at 1.20 and 1.22. Mean Aortic ISI: 0.22 ± 0.07, with the 5th and 10th percentiles at 0.13 and 0.14 and the 90th and 95th percentiles at 0.32 and 0.35. These additional indices provide a more comprehensive characterization of fetal aortic artery hemodynamics, enhancing the understanding of vascular function during pregnancy.

### 3.4. The Intraclass Correlation Coefficients (ICCs) for All Measurements

The ICC for PS was 0.993 (95% CI, 0.987–0.998), 0.946 (0.907–0.984) for ED, 0.709 (0.528–0.890) for VI, 0.849 (0.747–0.951) for PI, 0.744 (0.581–0.906) for RI, 0.997 (0.994–0.999) for TAMV, 0.794 (0.659–0.928) for IFI, and 0.987 (0.977–0.996) for ISI.

### 3.5. Relationship between Eight Doppler Indices and Gestational Age (Figure 3A–H)

In Figure 3A, where Y represents aortic PS and X represents gestational age, there is a noticeable increase in PS with advancing gestational age. Moving to Figure 3B, where Y represents aortic ED, the ED also exhibits an increase as gestational age progresses. Figure 3C demonstrates the relationship between Aortic S/D and gestational age. As gestational age increases, there is a corresponding rise in the S/D. In Figure 3D, representing aortic PI against gestational age, the PI shows an upward trend with increasing gestational age. Moving to Figure 3E, which depicts aortic RI against gestational age, the RI increases as gestational age advances. Figure 3F illustrates Aortic TAMV against gestational age, indicating an increase in TAMV with progressing gestational age. In Figure 3G, where Y represents Aortic IFI, the IFI of the aorta demonstrates a decreasing trend with gestational age. Finally, in Figure 3H, representing Aortic ISI against gestational age, the ISI decreases with advancing gestational age. These graphical representations provide a visual understanding of how these eight Doppler indices are associated with gestational age in the context of fetal aortic artery hemodynamics.

### 3.6. The Relationship between the Eight Doppler Indices and Maternal Age (MA), Nuchal Thickness (NT), Crown-Rump Length (CRL), and Fetal Heartbeat (FHB) (Figure 4)

A.MA:

There is no significant correlation between maternal age and any of the eight Doppler indices (PS, ED, S/D, PI, RI, TAMV, IFI, and ISI) (all *p* > 0.05).

B.NT:

Aortic PS and TAMV positively correlate with NT (*p* = 0.001, *p* = 0.000). Aortic ED, S/D, PI, RI, IFI, and ISI are not significantly correlated with NT (all *p* > 0.05).

C.CRL:

Aortic PS, ED, PI, and TAMV are positively correlated with CRL (*p* = 0.000, *p* = 0.003, *p* = 0.001, *p* = 0.000), and Aortic ISI is negatively correlated with CRL (*p* = 0.001). But there is no significant correlation between Aortic S/D, RI, IFI, and CRL (all *p* > 0.05).

D.FHB:

Aortic PS, PI, and TAMV negatively correlate with FHB. (*p* = 0.001, *p* = 0.005, and *p* = 0.026), Aortic ISI is positively correlated with FHB (*p* = 0.010). However, there is no significant correlation between Aortic ED, S/D, RI, IFI, and FHB (all *p* > 0.05).

Appendix A: Pearson correlations of birth weight (BW) and the Aortic PS, ED, Ns, ISI, IFI, NT, CRL, and MA in 24 live births.

## 4. Discussion

The conventional power Doppler technique for Doppler ultrasound utilizes a single color. When assessing the fetal aortic arch to identify the isthmus portion of the aorta, precise positioning and utilizing pulse Doppler in the traditional longitudinal aortic arch view (LAA view) can pose challenges. HDFI, on the other hand, assigns a color based on the measured Doppler shift without considering the direction or velocity [10]. Compared to Color Doppler flow images, HDFI exhibits superior resolution and sensitivity [11,12].

When evaluating the aortic isthmus, two sonographic planes are typically used: the traditional longitudinal aortic arch (LAA) view and the cross-sectional three vessels and trachea (3VT) view. The 3VT view is often considered quicker to obtain and less technically challenging to visualize, as it is commonly included in routine heart examinations. However, it lacks the simultaneous visualization of the head and neck vessels and relies on the waveform of the aortic isthmus for gate placement guidance, especially in the second trimester. In first-trimester scans, there is a preference for assessing from the LAA view due to concerns about the technical challenges associated with the 3VT view. This preference may stem from the fetus’s earlier gestational age and more active movements, which makes visualization difficult in the 3VT view [6].

In our current study, the four parameters (PS, PI, ED, and TAMV) that positively correlated with gestational age (all *p* < 0.01) among the six Doppler indices were assessed. Conversely, RI did not exhibit a significant positive correlation (*p* = 0.33), consistent with findings from previous studies that evaluated aortic isthmus blood flow in the second trimester [18,19,20]. As for the remaining two Doppler indices, IFI and ISI, their values decreased with increasing gestational age, but only ISI showed a significant negative correlation (*p* = 0.03). Unlike a prior study in the second trimester [21], IFI did not exhibit a significant correlation with gestational age in our study (*p* = 0.29). This disparity may be attributed to the following factors: one is the intra-observer variation, the second is a limitation of the study’s case numbers, and the third is the limited gestational period in the first trimester.

In addition to the findings above, we observed a correlation between the eight indices (Figure 3) and Crown-Rump Length (CRL), similar to that with gestational age. Aortic PS and TAMV were the only indices that exhibited a significant positive correlation with fetal nuchal translucency (NT) (*p* < 0.01). This result aligns with a previous study on fetal cardiac flow velocity and NT thickness [22]. Conversely, the Aortic isthmic systolic index (ISI) showed an inverse relationship with NT (*p* < 0.05), consistent with findings from another study [20,23].

When assessing the relationship between aortic Doppler indices and fetal heartbeat (FHB), Aortic PS, PI, and TAMV displayed a significant negative correlation with FHB (*p* < 0.01, *p* < 0.01, *p* < 0.05), consistent with previous research [24,25]. Similarly, as ISI is inversely proportional to PSV, it exhibited an inverse relationship with NT and CRL (*p* < 0.05) but a direct positive correlation with FHB (*p* < 0.05) (Figure 4).

In summary, Figure 4 shows that NT, CRL, and FHB are closely linked to the eight sonographic indices of the fetal aorta in the first trimester. Particularly, the PS, TAMV, and ISI indices are notable. PS and TAMV show a distinct positive correlation with NT and CRL, the two indicators utilized in first-trimester Down syndrome ultrasound screening tests, whereas ISI displays a negative correlation. However, ISI has a positive correlation with FHB. The potential of aortic PS and TAMV as supplementary soft markers for Down syndrome ultrasound screening in the first trimester warrants further validation through extensive large-scale studies.

For clinical purposes, previous studies have categorized five types of IFI [1]. Type I indicates an index higher than 1, indicating the presence of antegrade flow in both systole and diastole. This is typically observed in normal fetuses, with a progressive decrease in IFI throughout the second half of pregnancy. When comparing the Doppler waveforms of the aortic isthmus (AI) in normal fetuses between the first and second trimesters, the waveforms are generally similar, with one notable difference being the presence of a small prominent reverse spike, as illustrated in Figure 2.

In many published studies, the formation of the “incisura” in the aortic waveform typically occurs after 20 weeks of gestation [18,23,26]. However, our study found that a reverse spike (Figure 2) was detectable in most cases (detection rate: 95.4%), even during early gestation. This small prominent spike likely corresponds to the upper systolic nadir (Ns), consistently remaining above the baseline level (zero velocity line) between 11 and 14 weeks of gestation. These reverse spikes were consistently observed in nearly every waveform until the second trimester, after which the Ns gradually crossed the baseline level [20,23,27]. Despite knowing the Ns position accurately from the waveform, it is necessary to calculate the speed of Ns through interpolation to obtain the correct ISI.

In some cases, the left subclavian artery can serve as a guide for positioning the cursor below the color signal under power Doppler mode. The cursor must remain fixed without movement (i.e., the fetus must stay still). However, accurately assessing the position of the AI, especially in the early trimester, can be challenging. Doppler waveforms can aid in reconfirming whether the cursor is correctly positioned.

In our early trimester research, we observed an inverse relationship between ISI and gestational weeks (*p* < 0.05) (Figure 3H). Similarly, IFI also showed an inverse correlation with gestational age, although it did not reach statistical significance (Figure 3G). This finding is consistent with a previous report in the second trimester, which suggested a gradual decrease in IFI with gestational age [21].

Intrauterine growth restriction (IUGR) poses significant risks to fetal and neonatal health, elevating the likelihood of stillbirth and neonatal mortality. In obstetric care, there’s an increasing focus on employing new Doppler velocimetric parameters to identify and assess the severity of IUGR. In healthy fetuses, AI flow usually stays antegrade. However, in cases of IUGR, placental insufficiency can cause a gradual decrease in flow, often shifting towards predominantly retrograde flow. This change in signals compromises oxygen delivery to the fetal brain and might precede other indicators of severe hypoxemia detected by Doppler velocimetry in various vessels. The aortic Doppler holds promise as a valuable tool for managing pregnancies affected by IUGR. Its use could enhance clinical decision-making and potentially improve outcomes for both mother and baby [26].

Moreover, we conducted a follow-up on 27 cases of postnatal body weight (CRL less than 10%), with only three cases lost to follow-up. Of the 24 cases providing data for analysis, we observed notable correlations between fetal birth weight (BW) and MA, NT, and CRL [28]. Notably, NT [29,30] exhibited statistical significance (*p* < 0.05) (Appendix A).

Maternal characteristics, such as smoking and dyslipidemia, can significantly influence the early development of atherosclerosis. In one study analyzing maternal factors, including family history of cardiovascular disease, diabetes, and hypertension, researchers found that maternal smoking could already have a detrimental effect on the cardiovascular system of newborns. Similar effects were observed with gestational diabetes [31].

Our study boasts several strengths, notably using a pioneering ultrasonographic technique HDFI to establish reference values for fetal AI blood flow velocity and associated indices. During the first trimester, we identified significant correlations between Aortic PS, PI, TAMV, ISI, and NT, CRL, FHB. However, it is imperative to acknowledge the limitations inherent in our research, such as the relatively limited sample size, the absence of consideration of the potential influence in maternal characteristics, including smoking, familial hypertension, dyslipidemia, diabetes, cardiovascular disease, and other underlying disease, and the data was collected mainly from a single center.

In conclusion, our study marks the inaugural endeavor to set reference values for fetal aortic blood flow Doppler profiles, including ISI and IFI, within the 11 to 14 weeks of gestation age. Our findings underscore the potential of HDFI coupled with Doppler analysis as a novel ultrasound method for assessing the fetal aortic artery in the first trimester.

## Figures and Tables

**Figure 1 bioengineering-11-00378-f001:**
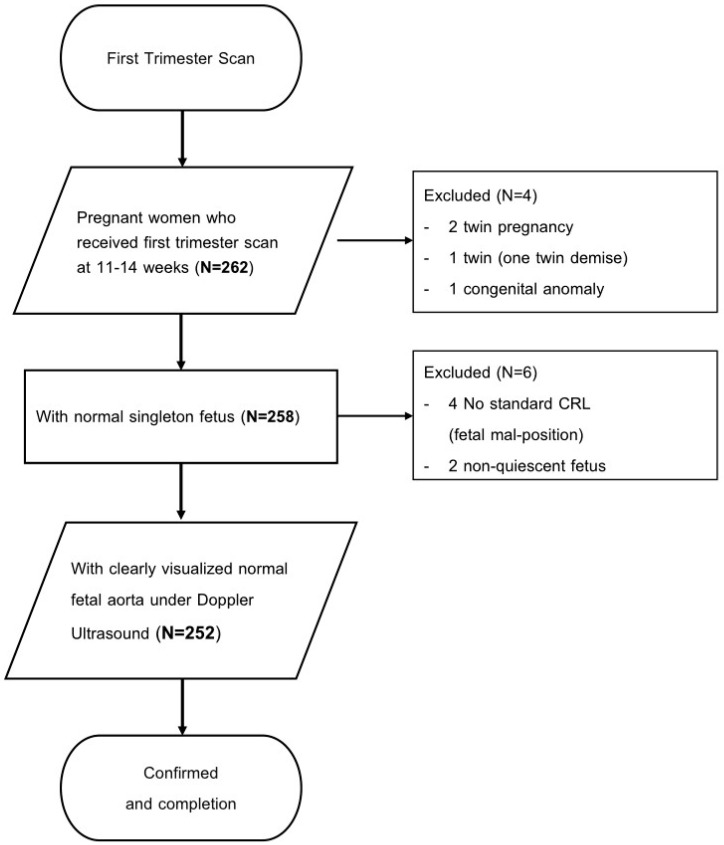
Study Flow Diagram of This Study.

**Figure 2 bioengineering-11-00378-f002:**
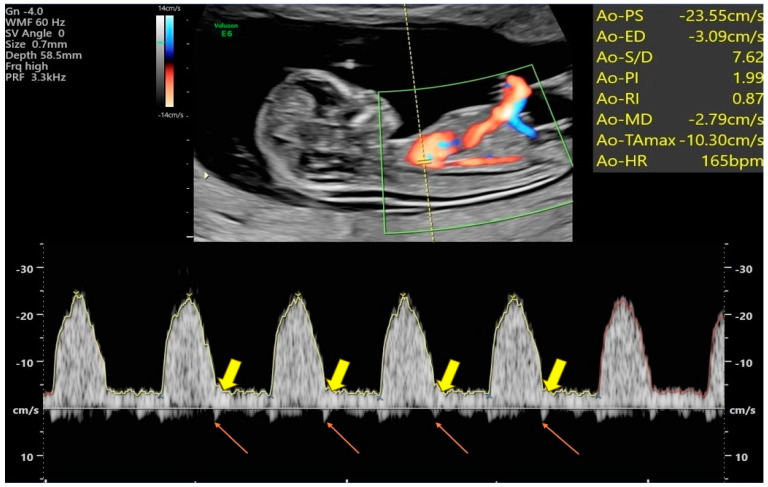
Power Doppler HDFI and aortic pulse Doppler spectral analysis were employed in a normal fetus at a gestational age of 12 + 1 weeks. The thin pinkish arrow indicates the reverse small prominent spike, while the thick yellowish arrow highlights the constant forward flow and possibly the “incisura”.

**Figure 3 bioengineering-11-00378-f003:**
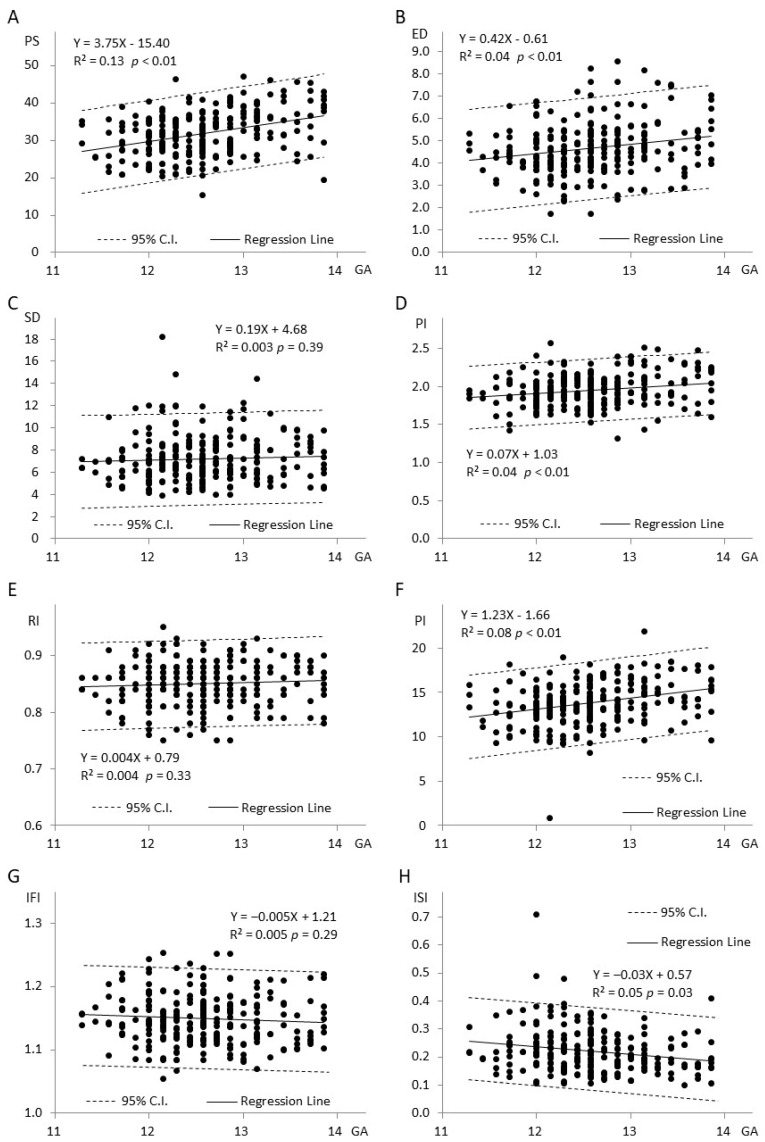
(**A**–**F**) Relationship between 6 Doppler indices of the fetal aortic artery and gestational age. (**G**) Relationship between IFI of the fetal aortic artery and gestational age. (**H**) Association between ISI of the fetal aortic artery and gestational age.

**Figure 4 bioengineering-11-00378-f004:**
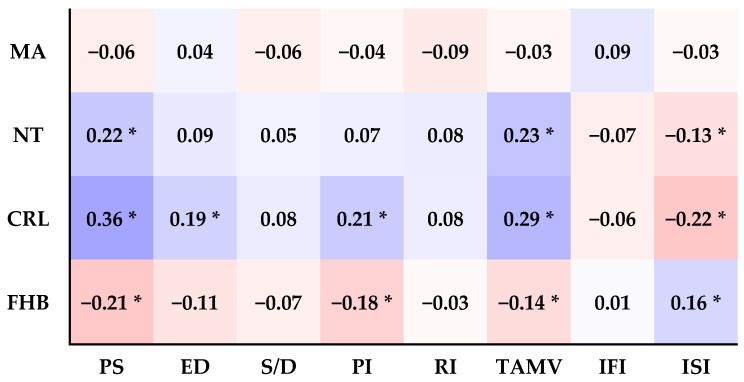
It illustrates the correlation between sonographic parameters, encompassing the fetal aortic artery’s six Doppler indices, IFI, and ISI, and clinical characteristics such as MA, NT, CRL, and FHB. Pearson’s correlation coefficient was utilized for this analysis. In the correlation matrix, negative correlation coefficient values are denoted in pink, while positive correlation coefficient values are represented in purple. A correlation coefficient value of zero is depicted in white. Significant correlations at the 0.05 level (two-tailed) are indicated by an asterisk (*).

**Table 1 bioengineering-11-00378-t001:** Essential characteristics of the general population.

No. of participants	252 cases
Maternal characteristics	Mean (SD) or %
Age (y/o)	29.19 (3.32)
Gestational Age (week)	12.57 (1.38)
Nulliparity	54%
Fetal characteristics	
Birth weight (g):	2908 (308)
>10th centile	89%
<10th centile	11%
Mode of delivery:	
Spontaneous	61%
Cesarean section	39%
Apgar score (1 min/5 min)	8/9

## Data Availability

The Excel data used to support this study’s findings were supplied by Yi-Cheng Wu under license. Requests for access to these data should be made to Yi-Cheng Wu (wu102007@gmail.com).

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
