# Peer review of "Fetal Aortic Blood Flow Velocity and Power Doppler Profiles in the First Trimester: A Comprehensive Study Using High-Definition Flow Imaging"

_bioengineering, 2024, doi:10.3390/bioengineering11040378_

Round 1
Reviewer 1 Report
Comments and Suggestions for Authors
To:
Editorial Board
Bioengineering
Title: “Fetal Aortic Blood Flow Velocity and Power Doppler Profiles in the First Trimester: A Comprehensive Study Using High-Definition Flow Imaging (HDFI)”
Dear Editor,
I read this paper and I think that:
- Inter and intraobserver variability coefficients should be provided in order to evaluate the reproducibility of the methods.
- What was the control test for these evaluations? Authors should provide data about this specific issue.
- The impact of several characteristics of the maternal population should be provided as they can impact on vascular performances of the foetuses and future births. Please update data.
- Similarly, authors can discuss the paper from Ciccone MM et al. Biomed Res Int. 2013;2013:459168.
- How many alterations in morphology of the foetuses were? Please include a flow chart of the paper.
- Were the foetuses consecutively enrolled? Please specify.
- How might authors standardize the differences in flow velocity?
Author Response
Comments and Suggestions from Reviewer #1:
Comment 1:
Inter and intra-observer variability coefficients should be provided in order to evaluate the reproducibility of the methods.
Response 1:
Thank you for your precious comments and suggestions.
We very much agree with your comments. However, the Kappa value can only be calculated if different sonographers examine the same patient or if the same sonographer examines the same patient twice. One of the authors (Y.C.W.) performs all the examinations. The intraclass correlation coefficients (ICCs) for all measurements: the ICC for PS was 0.993 (95% CI, 0.987–0.998), 0.946 (0.907–0.984) for ED, 0.709 (0.528–0.890) for VI, 0.849 (0.747–0.951) for PI, 0.744 (0.581–0.906) for RI, 0.997 (0.994–0.999) for TAMV, 0.794 (0.659-0.928) for IFI, and 0.987 (0.977-0.996) for ISI. (Page 7, Line 182-185).
Comment 2:
What was the control test for these evaluations? Authors should provide data about this specific issue.
Response 2:
Thank you for your precious comments.
This study is not a case-control study, so we do not have the control test for these evaluations.
Comment 3:
The impact of several characteristics of the maternal population should be provided as they can impact on vascular performances of the foetuses and future births. Please update data.
Response 3:
Thank you for your precious comments.
- We added maternal characteristics to our discussion and updated our data in the following paragraph:
Maternal characteristics, such as smoking and dyslipidemia, can significantly influence the early development of atherosclerosis. In one study analyzing maternal factors, including family history of cardiovascular disease, diabetes, and hypertension, researchers found that maternal smoking could already have a detrimental effect on the cardiovascular system of newborns. Similar effects were observed with gestational diabetes31.
(Please check Page 14, Line 333-337)
- We also add one reference in 31 for the above paragraph.
31.Ciccone MM et al., Biomed Res Int. 2013;2013:459168
- We admitted to the absence of consideration of maternal characteristics and thought as a shortcoming in our manuscript, so we have added some revised paragraphs:
limited small sample size, the absence of consideration of the potential influence of maternal characteristics, including smoking, familial hypertension, dyslipidemia, diabetes, cardiovascular disease, and other underlying disease, and the collection of data collected mainly from a single center.
(Please check Page 14, Lines 343-346)
Comment 4:
Similarly, authors can discuss the paper from Ciccone MM et al. Biomed Res Int. 2013;2013:459168.
Response 4:
Thank you for your precious comments.
We have added some paragraphs to the Discussion of the manuscript.
(Please check Page 14, Line 333-337, 343-346)
Comment 5:
How many alterations in morphology of the foetuses were? Please include a flow chart of the paper.
Response 5:
Thank you for your precious comments.
- Fetuses have different morphologic alterations related to their growth during 11-14 weeks of gestation. As these vary case by case, we cannot know the number of morphologic alterations for the fetuses examined in this study.
- We have added a study flowchart in Figure 1.
Comment 6:
Were the foetuses consecutively enrolled? Please specify.
Response 6:
Thank you for your precious comments.
Yes, the fetuses are consecutively enrolled as their mothers enter our center for the first trimester Down syndrome scan at 11-14 weeks of gestation, which corresponds with our inclusion criteria for this study.
Comment 7:
How might authors standardize the differences in flow velocity?
Response 7:
Thank you for your precious comments.
The differences were standardized below procedures.
- Angle Correction:
Doppler ultrasound measures blood flow velocity based on the angle between the direction of blood flow and the ultrasound beam. An angle correction is applied to the measured velocity to account for this angle. When the ultrasound beam is parallel to the direction of blood flow, the measured velocity is accurate. However, it needs correction when the angle deviates from 0 degrees.
----------> In our study, the angle between the Doppler vernier and the assumed direction of blood flow was always kept at
0 degree. (on Page 5, Lines 110-111)
- Sample Volume Placement:
The Doppler sample volume should be placed accurately within the vessel, representing the area from which the velocity data is collected. Placing it in the center of the vessel helps ensure accurate measurements.
----------> In our study, we pressed Doppler mode (as shown in Fig 1) and placed the sample volume in an optimal location adjacent to the access to the aortic arch. The sample volume gap was kept less than or equal to 1 mm. (on Page 5, Lines 107-110)
- Consistent Depth:
Doppler ultrasound measures blood flow velocity at a specific depth within the vessel. Keeping the depth consistent during measurements helps maintain standardization.
----------> In our study, we used this innovative sonographic technique of high-definition flow imaging (HDFI) to keep consistent depth and maintain standard measurement. (on Page 5, Lines 108-110)
- Calibration:
Regular calibration of the Doppler system ensures accurate velocity measurements. Calibration involves comparing Doppler velocity measurements with a known standard and adjusting the system settings if necessary.
----------> In our study, our ultrasound equipment has signed a maintenance contract with the manufacturer, and specialized technicians will regularly calibrate and maintain our equipment.
Use of Standard Protocols: Following standardized protocols for Doppler ultrasound measurements helps ensure consistency and accuracy across different operators and facilities.
----------> In our study, following adjustments to the power Doppler Angio mode (PRF: 1.8 kHz, WMF: low 1), detectable Doppler signals, defined as reproducible, similar arterial waveforms obtained for at least three consecutive cardiac cycles, were analyzed. (on Page 5, Lines 117-119)
- Quality Control:
Regular quality control checks, including phantom studies and inter-operator variability assessments, help identify and correct any discrepancies in velocity measurements.
By applying these techniques, Doppler ultrasound measurements can be standardized to account for differences in flow velocity and improve the accuracy and reliability of blood flow velocity assessments.

Reviewer 2 Report
Comments and Suggestions for Authors
General Comments
These authors performed a cross-sectional clinical study aimed at measuring fetal aortic blood flow velocity and associated indices in the first trimester. To this aim, they used an innovative ultrasonographic technique called high-definition flow imaging (HDFI) to measure parameters related to the blood velocity in the aortic isthmus of 252 fetuses with gestational ages ranging from 11 to 14 weeks.
This study has found appealing results regarding the development of fetal aortic isthmus blood flow as early as the first trimester of fetus life. However, the manuscript is weakened by a non-adequate format and results presentation and therefore needs further improvements.
Major Comments
My main concern is the format of the manuscript. Although it is rich of novel and important data findings, the manuscript needs improvements towards a good standard of publication of clinical results from controlled clinical studies. Regarding this aspect, I highly advise the authors to check with the Journal and/or MDPI publisher if it is required that authors must observe the specifications for reports of certain study types, e.g. the CONSORT checklist for reports of randomized controlled trials. Or alternatively I can also suggest the ICMJE recommendations for the conduct, reporting, and publication of research work in Medical Journals.
Another great concern is the use of acronyms because these are constantly re-defined in the main text, and this is very annoying for the readership. Acronyms or abbreviations should be defined only once, at the first occurrence in text and then always used throughout the manuscript or in figures and tables. If there are too many, a list of abbreviations at the end of the manuscript might help the readership. Special cases are the Title (HDFI?) and the Abstract section, please check with the Journal.
Consider changing Figure 1 with one in vectorial format that does not stretch text and that may comply with the CONSORT format. Do not write “Study flow diagram” on image, it’s in the figure caption.
Figure 4 is introduced at page 7 and then is evoked in the Discussion but is not presented in the Results section. In general, figures/tables with actual study results should be listed in order of appearance and presented in the Results section. Further statements on the results may be written in the Discussion section when highlighting the study findings.
Supplemental Table “Supple 1” should not reside in the main text: check with the Journal for the rules regarding Supplemental materials. Define what N is. Also, tables cannot have vertical lines.
Minor Comments
1) Page 2, line 84: define what IVF is.
2) Check definition of CRL; at line 78 is crown-rump length, at line 92 is fetal body length, whereas at line 101 is cranial rump length.
3) Evaluate the phrases between lines 102 and 110, especially for the use of verbs. The same for the text between lines 164 to 167.
4) Section 3.4: do not repeat what X axis is.
5) Check the definition PS, sometimes is also PSV (line 118).
6) Figure 3: consider providing a unique legend regarding 95% CI and regression line for all plots; revise these lines for plots G and H.
Comments on the Quality of English LanguageEnglish must be carefuly reviewed.
Author Response
Major Comments:
Comment 1:
My main concern is the format of the manuscript. Although it is rich of novel and important data findings, the manuscript needs improvements towards a good standard of publication of clinical results from controlled clinical studies. Regarding this aspect, I highly advise the authors to check with the Journal and/or MDPI publisher if it is required that authors must observe the specifications for reports of certain study types, e.g. the CONSORT checklist for reports of randomized controlled trials. Or alternatively I can also suggest the ICMJE recommendations for the conduct, reporting, and publication of research work in Medical Journals.
Our Responses:
Thank you for your precious comments.
- We did not find the CONSORT checklist among the five considerations in the Submission Checklist from (https://www.mdpi.com/journal/data/instructions). Therefore, we followed the Reviewer’s suggestion to adhere to the ICMJE criteria: https://www.icmje.org/recommendations/browse/publishing-and-editorial-issues/clinical-trial-registration.html.
- The attached file is our disclosure form for the ICMJE.
Comment 2:
Another great concern is the use of acronyms because these are constantly re-defined in the main text, and this is very annoying for the readership. Acronyms or abbreviations should be defined only once, at the first occurrence in text and then always used throughout the manuscript or in figures and tables. If there are too many, a list of abbreviations at the end of the manuscript might help the readership. Special cases are the Title (HDFI?) and the Abstract section, please check with the Journal.
Our Responses:
Thank you for your precious comments.
- We thank the Reviewer for his comment. We have revised the manuscript to use the same abbreviations in the text, figures, and tables.
- We delete the (HDFI) in the title of our topics.
- In the abstract:
Line 16: high-definitions flow imaging (HDFI), delete HD flow imaging.
Line 67-68: HDFI
Line 72: HDFI
Line 75-76: HDFI
In the Discussion:
Line 247: HDFI
Line 249: HDFI
Line 308: delete “HD flow.”
Line 349-350: delete "power Doppler angiography with High-Definition Flow Imaging.”
Comment 3:
Consider changing Figure 1 with one in vectorial format that does not stretch text and that may comply with the CONSORT format. Do not write “Study flow diagram” on image, it’s in the figure caption.
Our Responses:
Thank you for your precious comments.
As suggested, we have changed Figure 1 to a vectorial format (Page 4, Figure 1).
Comment 4:
Figure 4 is introduced at page 7 and then is evoked in the Discussion but is not presented in the Results section. In general, figures/tables with actual study results should be listed in order of appearance and presented in the Results section. Further statements on the results may be written in the Discussion section when highlighting the study findings.
Our Responses:
Thank you for your precious comments.
- We have added the following paragraphs to the results.
- We added a paragraph about the finding in Figure 4 to the result from Pages 7 and 8, Lines 203-219.
3.6. The relationship between the eight Doppler indices and maternal age, nuchal thickness (NT), crown-rump length (CRL), and fetal heartbeat (FHB) (Figure 4)
- Maternal Age:
There is no significant correlation between maternal age and any of the eight Doppler indices (PS, ED, S/D, PI, RI, TA max, IFI, and ISI) (all P > 0.05).
- Nuchal Thickness (NT):
Ao PS and Ao TA max positively correlate with NT (p = 0.001, p = 0.000). Ao ED, Ao S/D, Ao PI, Ao RI, Ao IFI, and Ao ISI are not significantly correlated with NT (all P > 0.05).
- Crown-Rump Length (CRL):
Ao PS, Ao ED, Ao PI, and Ao TA max are positively correlated with CRL (p=0.000, p=0.003, p=0.001, p=0.000), and Ao ISI is negatively correlated with CRL (p=0.001). But there is no significant correlation between Ao S/D, Ao RI, Ao IFI, and CRL (all P>0.05)
- Fetal Heartbeat (FHB):
Ao PS, Ao PI, and Ao TA max negatively correlate with FHB. (p=0.001, p=0.005, and p=0.026), Ao ISI is positively correlated with FHB.(p=0.010). However, there is no significant correlation between Ao ED, Ao S/D, Ao RI, Ao IFI, and FHB. (all P>0.05).
- We also have added a paragraph about Figure 4 in the discussion section. In Page 13, Line 283-290.
Comment 5:
Supplemental Table “Supple 1” should not reside in the main text: check with the Journal for the rules regarding Supplemental materials. Define what N is. Also, tables cannot have vertical lines.
Our Responses:
Thank you for your precious comments.
- We have removed it from the main text.
- “N” denotes the quantity of cases.
- We have revised the Supplement 1 table as follows,
|
|
||||||||||||
|
Supplement 1: |
|
|||||||||||
|
Pearson Correlations of birth weight and the Ao (PS), Ns, ISI, IFI, NT, CRL, maternal age in 24 live births. |
||||||||||||
|
|
||||||||||||
|
  |
BW |
ISI |
Ns |
PS |
ED |
PS+ED |
IFI |
AGE |
NT |
CRL |
GP (%) |
|
|
BW |
1 |
0.023 |
0.114 |
0.118 |
-0.025 |
0.098 |
-0.110 |
0.383 |
0.431* |
0.395 |
-0.251 |
|
|
ISI |
1 |
0.677** |
-0.508* |
-0.327 |
-0.508* |
-0.035 |
-0.363 |
0.222 |
-0.229 |
-0.413* |
|
|
|
Ns |
1 |
0.242 |
0.168 |
0.245 |
-0.096 |
-0.283 |
0.337 |
-0.001 |
-0.137 |
|
||
|
PS |
1 |
0.575** |
0.986** |
-0.147 |
0.179 |
0.147 |
0.364 |
0.381 |
|
|||
|
ED |
1 |
0.704** |
0.713** |
-0.031 |
0.200 |
0.317 |
0.359 |
|
||||
|
PS+ED |
1 |
0.019 |
0.149 |
0.168 |
0.381 |
0.405* |
|
|||||
|
IFI |
1 |
-0.135 |
0.109 |
0.100 |
0.134 |
|
||||||
|
AGE |
1 |
0.090 |
0.177 |
0.065 |
|
|||||||
|
NT |
1 |
0.353 |
0.230 |
|
||||||||
|
CRL |
1 |
0.183 |
|
|||||||||
|
GP (%) |
  |
  |
  |
  |
  |
  |
  |
  |
  |
  |
1 |
|
|
*. Correlation is significant at the 0.05 level (2-tailed). GP (%): Growth Percentile |
||||||||||||
|
**. Correlation is significant at the 0.01 level (2-tailed). |
||||||||||||
Minor Comments
Comment 1:
- Page 2, line 84: define what IVF is.
Our Responses:
Thank you for your precious comments.
- Taiwan IVF Group is the name of a clinic for Reproductive Medicine & Infertility in Hsinchu, Taiwan.
- Taiwan IVF Group is one of the affinities of our first author (Dr. Yi-Cheng, Wu).
Comment 2:
- Check definition of CRL; at line 78 is crown-rump length, at line 92 is fetal body length, whereas at line 101 is cranial rump length.
Our Responses:
Thank you for your precious comments.
- The definition of CRL is the abbreviation of Crown-Rump Length.
- Line 93: revised as fetal CRL.
- Line 102-103: revised while measuring the fetal CRL in a lying-up posture.
Comment 3:
3) Evaluate the phrases between lines 102 and 110, especially for the use of verbs. The same for the text between lines 164 to 167.
Our Responses:
Thank you for your precious comments.
Line 102-114
We revised the following sentences:
First, we obtained a clear longitudinal section of the fetal aortic arch while measuring the fetal CRL in a lying-up posture. Then, we pressed the power Doppler mode to make the signal appear on the entire aorta while waiting for the fetus to be entirely still. Then, we pressed Doppler mode (as shown in Fig 1) and placed the sample volume in an optimal location adjacent to the access to the aortic arch. The sample volume gap was kept less than or equal to 1 mm. The angle between the Doppler vernier and the assumed direction of blood flow was always kept below 15 degrees. After obtaining at least three consecutive uniform waveforms, we froze the images and stored them for subsequent data analysis. We executed and saved all records, ensuring that Doppler ultrasound waveforms of the aorta were obtained in the same fetal position and posture.
Line 165-168
We revised the following sentences:
The mean of the aortic time-averaged maximum velocity (TA max) was 13.79 ± 2.59 cm/sec. The 5th and 10th percentiles were 9.57 cm/sec and 10.63 cm/sec, while the 90th and 95th percentiles were 16.95 cm/sec and 17.84 cm/sec.
Comment 4:
- Section 3.4: do not repeat what X axis is.
Our Responses:
Thank you for your precious comments.
Line 189-197
We revised the following sentences:
In Figure 3(A), where Y represents aortic PS and X represents gestational age, there is a noticeable increase in PS with advancing gestational age. Moving to Figure 3(B), where Y represents aortic ED, the ED also increases as gestational age progresses. Figure 3(C) demonstrates the relationship between Aortic S/D and gestational age. As gestational age increases, there is a corresponding rise in the S/D. In Figure 3(D), representing Aortic PI against gestational age, the PI shows an upward trend with increasing gestational age. Moving to Figure 3(E), which depicts aortic RI against gestational age, the RI increases as gestational age advances. Figure 3(F) illustrates Aortic TAMV against gestational age, indicating an increase in TAMV with progressing gestational age. In Figure 3(G), where Y represents Aortic IFI, the IFI of the aorta demonstrates a decreasing trend with gestational age.
Comment 5:
- Check the definition PS, sometimes is also PSV (line 118).
Our Responses:
Thank you for your precious comments.
The definition of the PS is the peak systolic velocity.
We revised the following sentences:
Line 123
(IFI)=PS+ED/PS and Isthmic systolic index (ISI) = Ns (systolic nadir)/PS were also assessed, respectively.
Comment 6:
6) Figure 3: consider providing a unique legend regarding 95% CI and regression line for all plots; revise these lines for plots G and H.
Our Responses:
Thank you for your precious comments.
We revised Figure 3 and made a new figure on Page 10.

Round 2
Reviewer 1 Report
Comments and Suggestions for Authors
Authors well addressed my previous comments. The paper improved very much
Author Response
Dear Reviewer:
We are truly grateful for the reviewers’ constructive comments, which improved this manuscript.
Reviewer 2 Report
Comments and Suggestions for Authors
General Comments
These authors did not perform a thorough revision of the original manuscript, the paper still needs further refinement.
Major Comments
Although the authors declared to adhere to the ICMJE criteria for publication of clinical trials results, the actual manuscript lacks information of clinical trial registration. Briefly, the ICMJE requires that all medical journal editors require registration of clinical trials in a public trials’ REGISTRY at or before the time of first patient enrollment as a condition of consideration for publication.
If your study protocol was already registered, please provide this info – including registry code - in the Methods section. Otherwise provide at least evidence from your Institution’s Ethical Committee and/or IRB number.
Another great concern remains the use of abbreviations for clinical parameters in the manuscript, please see below.
Specific Comments
Regarding abbreviations, once defined, do not use them between parentheses (). Another annoying issue is the use of word “aortic”: decide only one among “Aortic”, “aortic”, or “Ao” and use it consequently, otherwise it’s a mess.
Below I corrected only the Abstract, but you should carefully check the whole text and figure legends, tables, and Supplemental Material. Even though acronyms were introduced first in the Abstract, at the first occurrence in the remaining text should be re-defined again.
From the Abstract:
“Utilizing 2D power Doppler ultrasound images, aortic blood flow parameters were assessed, including aortic peak systolic velocity (PS), aortic end-diastolic velocity (ED), aortic time average maximal velocity (TAMV), and various indices such as aortic S/D (systolic velocity/diastolic velocity), aortic PI (pulsatile index), aortic RI (resistance index), aortic isthmus flow velocity index (IFI), and aortic isthmic systolic index (ISI).”
should be:
“Utilizing 2D power Doppler ultrasound images, aortic blood flow parameters were assessed, including aortic peak systolic velocity (PS), aortic end-diastolic velocity (ED), aortic time average maximal velocity (TAMV), and various indices such as aortic systolic velocity/diastolic velocity S/D, aortic pulsatile index (PI), aortic resistance index (RI), aortic isthmus flow velocity index (IFI), and aortic isthmic systolic index (ISI).”
“...a positive correlation between gestational age and PS...” should be: “...a positive correlation between gestational age (GA) and PS...”.
“Notably, aortic (PS), aortic (TAMV), and aortic (ISI) exhibit significant correlations with NT, CRL, and Fetal Heartbeat, with (ISI) appearing more relevant than (IFI), PS, TAMV, and fetal heartbeat.”
should be:
“Notably, aortic PS, TAMV, and ISI exhibit significant correlations with NT, CRL, and FHB, with ISI appearing more relevant than IFI [check this!], PS, TAMV, and FHB [check this!].”
Also, define additional keywords, at least six.
Minor Comments
1) All section 3.6: TAmax should be TAMV.
2) TA on Figure 4 should be TAMV.
Should be improved.
Author Response
MDPI
Bioengineering Editorial Office
St. Alban-Anlage 66, 4052 Basel, Switzerland
Tel.: +41 61 683 77 34
Fax: +41 61 302 89 18
Re: MS#: Bioengineering-2901850
Dear Editor:
Thank you for your email regarding our submitted manuscript on March 29, 2024 (MS#: diagnostics-2901850). We are truly grateful for the reviewers’ constructive comments, which improved this manuscript. Please find below our point-by-point responses to the Reviewers’ comments/questions. We hope our responses and the revised manuscript meet the journal’s standards and are accepted for publication.
Thank you very much for your assistance.
Sincerely yours,
Woeichyn Chu, Ph.D.
Distinguished Professor,
Department of Biomedical Engineering,
National Yangming Chiaotung University
Address No. 155, Section 2, Linong Street, Beitou District, Taipei City.
Email: wchu@nycu.edu.tw
Telephone 886-2-28267025 Lab. Telephone 886-2-28267000, ext.5497
General Comments
These authors did not perform a thorough revision of the original manuscript, the paper still needs further refinement.
Our Responses:
We thank the Reviewer’s precious Comments.
We have thoroughly revised the original manuscript and further refined our paper.
Major Comments:
Although the authors declared to adhere to the ICMJE criteria for publication of clinical trials results, the actual manuscript lacks information of clinical trial registration. Briefly, the ICMJE requires that all medical journal editors require registration of clinical trials in a public trials’ REGISTRY at or before the time of first patient enrollment as a condition of consideration for publication.
If your study protocol was already registered, please provide this info – including registry code - in the Methods section. Otherwise provide at least evidence from your Institution’s Ethical Committee and/or IRB number.
Our Responses:
We thank the Reviewer’s precious Comments.
- This study was approved by the Medical Ethics and Institutional Review Board of the Taoyuan General Hospital, Ministry of Health and Welfare. (IRB No: TYGH 112002)
- We have provided the Institutional Review Board Statement in the Materials and Methods. (Page 2, Lines 84-85)
Specific Comments
Regarding abbreviations, once defined, do not use them between parentheses (). Another annoying issue is the use of word “aortic”: decide only one among “Aortic”, “aortic”, or “Ao” and use it consequently, otherwise it’s a mess.
Our Responses:
We thank the Reviewer’s precious Comments. We have revised the manuscript accordingly.
We have revised the following: Lines 61, 74, 79-80, 130, 163, 166, 169, 171, 173, 175, 206, 212-213, 215-217, 219-221, 235 (Fig 4), 241-242, Supplement Table 1, 274-277, 280-282, 297, 312, 322, 331-333, 340-342.
Below I corrected only the Abstract, but you should carefully check the whole text and figure legends, tables, and Supplemental Material. Even though acronyms were introduced first in the Abstract, at the first occurrence in the remaining text should be re-defined again.
From the Abstract:
“Utilizing 2D power Doppler ultrasound images, aortic blood flow parameters were assessed, including aortic peak systolic velocity (PS), aortic end-diastolic velocity (ED), aortic time average maximal velocity (TAMV), and various indices such as aortic S/D (systolic velocity/diastolic velocity), aortic PI (pulsatile index), aortic RI (resistance index), aortic isthmus flow velocity index (IFI), and aortic isthmic systolic index (ISI).”
- According to the Reviewer’s suggestions,
In “Utilizing 2D power Doppler ultrasound images…., aortic isthmus flow velocity index (IFI), and aortic isthmic systolic index (ISI).”
Our Responses:
We thank the Reviewer’s precious comments.
We have revised it as “aortic systolic velocity/diastolic velocity (S/D), aortic pulsatile index (PI), aortic resistance index (RI), aortic isthmus flow” in Lines 22-23.
- In“...a positive correlation between gestational age and PS...” should be:
“...a positive correlation between gestational age (GA) and PS...”.
Our Responses:
We thank the Reviewer’s precious comments.
We have revised it as a positive correlation between gestational age (GA) and PS...” in Lines 26.
- In“Notably, aortic (PS), aortic (TAMV), and aortic (ISI) exhibit significant
correlations with NT, CRL, and Fetal Heartbeat, with (ISI) appearing more
relevant than (IFI), PS, TAMV, and fetal heartbeat.”
should be:
“Notably, aortic PS, TAMV, and ISI exhibit significant correlations with NT, CRL, and FHB, with ISI appearing more relevant than IFI [check this!], PS, TAMV, and FHB [check this!].”
Our Responses:
We thank the Reviewer’s precious comments.
We have revised it in Lines 38-40.
Notably, aortic PS, TAMV, and ISI exhibit significant correlations with NT, CRL, and FHB, with ISI appearing more relevant than IFI, PS, TAMV, and FHB.
- Also, define additional keywords, at least six.
Our Responses:
We thank the Reviewer’s precious comments.
We have revised our keywords as
Keywords: Power Doppler ultrasound; high-definition flow imaging (HDFI); aortic isthmus; isthmic flow index; isthmic systolic index; first-trimester screening; fetus
In Lines 43-44.
Minor Comments
Our Responses:
We thank the Reviewer’s precious comments.
We have revised it according to your suggestions as follows:
- All section 3.6: TAmax should be TAMV. In Line: 210, 212, 215,219.
- TA in Figure 4 should be TAMV. In Figure 4.

Round 3
Reviewer 2 Report
Comments and Suggestions for Authors
The authors addressed satisfactorily all my comments.